# A Feasible Methodological Approach to Estimate the Burden of Autism Spectrum Disorder: Results from the EPI-ASD Study in the Province of Lecce (Southern Italy)

**DOI:** 10.3390/ijerph19106334

**Published:** 2022-05-23

**Authors:** Giovanni Imbriani, Tiziana Grassi, Francesco Bagordo, Giovanni De Filippis, Donato De Giorgi, Luigi Peccarisi, Federica Dileone, Tonia Fattizzo, Gianfranco Antonucci, Maria Luciana Margiotta, Serafino De Giorgi, Valeria Grasso, Antonella De Donno, Prisco Piscitelli

**Affiliations:** 1Department of Biological and Environmental Science and Technology, University of Salento, 73100 Lecce, Italy; giovanni.imbriani@unisalento.it (G.I.); tiziana.grassi@unisalento.it (T.G.); antonella.dedonno@unisalento.it (A.D.D.); 2Medical Professional Association of Lecce Province (OMCEO Lecce), 73100 Lecce, Italy; giov.defilippis@gmail.com (G.D.F.); donatodg@libero.it (D.D.G.); ginopeccarisi@libero.it (L.P.); cepsia@ausl.le.it (G.A.); polo1.npi@ausl.le.it (M.L.M.); direzione.dsm@ausl.le.it (S.D.G.); priscofreedom@hotmail.com (P.P.); 3Local Health Authority of Lecce (ASL/LE), Department of Mental Health, 73100 Lecce, Italy; federicadileone@gmail.com (F.D.); toniafattizzo@libero.it (T.F.); 4Italian Ministry of Health, 00153 Rome, Italy; valeriagrasso.v@gmail.com

**Keywords:** autism spectrum disorder, epidemiology, prevalence, diagnosis, school

## Abstract

Diagnoses of Autism Spectrum Disorder (ASD) have rapidly increased globally. However, the lack of comprehensive epidemiological surveys and surveillance systems, able to provide official data at a national or European level is one of the main issues in the monitoring of this condition. The present study aimed to estimate the prevalence of ASD in children and adolescents aged 3–18 years old living in the province of Lecce (Southern Italy) through official data provided by the Local Health Authority of Lecce (ASL/LE) up to 31 October 2020, and compare it with school-based data concerning the number of students needing support for ASD. Based on data provided by the ASL/LE, in 2020 there were 509 cases of ASD among children and adolescents aged 3–18 years old, corresponding to a prevalence of 0.46%. A total of 408 (80.2%) were boys and 101 (19.8%) were girls. In relation to their age, 155 ASD cases (0.90%) were diagnosed in the 3–5 age group, while 222 (0.55%) in the 6–11 age group and 132 (0.25%) in the 12–18 age group. Prevalence of ASD assessed by school-based dataset was underestimated in the 3–5 age group, while the 6–11 and 12–18 age groups were consistent with the official data provided by the ASL/LE.

## 1. Introduction

Autism Spectrum Disorder (ASD) is a set of heterogeneous conditions affecting neuro-logical development characterized by early difficulties in social communication and repetitive behaviors associated with limited interests in usual childhood activities [1]. At the basis of its complex etiology, a genetic component has been identified by the identification of specific mutations in several high-confidence genes involved in the neuronal and cortical organization, in the formation and maturation of brain synapses, and neuro-transmission [2,3,4,5] that can lead to transient interruptions of brain development processes [6].

As found in epidemiological studies conducted in recent years, diagnoses of ASD have rapidly increased globally. According to the Autism and Developmental Disabilities Monitoring Network (ADDM), a group of programs funded by the American Centers for Disease Control and Prevention (CDC), the prevalence of ASD in the United States among children aged 8 years old has increased constantly passing from 1 out of 150 (0.66%) in the year 2000, to 1 out of 44 (2.27%) in the year 2018 [7,8,9,10].

This epidemiological phenomenon could be in part explained by the increase in awareness of early ASD signs and symptoms, as well as the improvements in the access to proper diagnosis through the availability of healthcare facilities and well characterized diagnostic criteria [11,12,13]. However, it is difficult to compare the prevalence rates across time and in different countries to highlight temporal or geographical variations as there are huge differences in methodologies and assessment tools used in each survey, as well as some variability in the age groups and size of the population studied [14]. Moreover, the prevalence assessment has often been performed within cross-sectional studies or in the frame of research projects rather than in standardized and comprehensive national surveillance programs.

In Europe, the estimated prevalence of ASD among children aged 7 to 9 in 2015 was 1.24% in Denmark, 0.76% in Finland, and 2.68% in Iceland. In France, the prevalence of ASD in the same year was assessed at 0.73% in the South West and 0.48% in the South East of France [15].

In Italy, the epidemiological data about ASD based on validated data sources are very poor and a national register is far from being created, so the evaluation of ASD prevalence and the evolution of the phenomenon over time is very difficult to carry out. Limited data are available at a local level. Overall, the main data sources are represented by the Territorial Centres for Autism (CAT) provided by all the Local Health Authorities of the 105 Italian provinces over 20 regions, where children with suspected ASD are subject to final diagnosis according to the criteria described in the Diagnostic and Statistical Manual of Mental Disorders, 5th edition (DSM-5) [16], and subsequently followed up. Additional data sources are represented by the School Offices set in each province (USP) and region (USR), as the Italian law no. 104/1992 guarantees the right to education by attending ordinary schools for children with disabilities or emotional/behavioral difficulties from up to the age of 18. Therefore, children with special educational needs, including those with ASD—after having been officially diagnosed by Local Health Authorities and provided with a specific certification indicating the diagnosis and the level of disability—receive adequate support including a dedicated teacher. Based on school data, in the province of Pisa, a study was conducted on schoolchildren aged 7–9 years old as a part of a European project (“Autism Spectrum Disorders in the European Union”, ASDEU), showing a prevalence of 1 out of 87 children (1.14%) in the age group 7–9 years old, needing school support for ASD [17]. Another Italian study carried out in the Emilia Romagna region assessed the trend in diagnoses of ASD among the population from 0 to 17 years old followed up at mental health services within the region from 2016 to 2019. In 2016, the prevalence was 0.38%, while in 2017 it was calculated as 0.45%, and in 2018 it reached 0.52%, with a peak of 0.6% in 2019 [18].

The present study aimed to estimate the prevalence of ASD in children and adolescents aged 3–18 years old living in the province of Lecce (Southern Italy) through official data provided by the Local Health Authority of Lecce (ASL/LE) up to 31 October 2020, and compare these findings with school-based dataset concerning the number of students individually supported with a dedicated teacher due to diagnosis of ASD.

## 2. Materials and Methods

### 2.1. Study Design

This study presents the first results of the “Autism Spectrum Disorder Epidemiology and Environmental Factors” (EPI-ASD) project, carried out in 2020–2021 by the University of Salento in cooperation with ASL/LE. It was an observational epidemiological study that aimed at (i) assessing the epidemiology of ASD in the province of Lecce by using official data maintained at Territorial Centers for ASD of the ASL/LE, where all the subjects with ASD living in the province are diagnosed and followed up, as well as other data sources, such as school-based dataset. This latter indirect source of data resumes the number of individual teachers granted by each school to support students with a diagnosis of ASD once officially confirmed by the ASL/LE; (ii) investigating the possible association between ASD and individual or family factors occurring in recruited children and their parents, through a retrospective georeferenced case-control study.

### 2.2. Study Area

The province of Lecce (2799.07 km^2^) corresponds to the Southern part of the Salento Peninsula, located in the Apulia Region (Southern Italy), and includes 96 municipalities with a total population of 782,165 inhabitants.

The health needs of people living in the province of Lecce are met by the ASL/LE which has the task of managing and coordinating public social and health services as well as the prevention and control of infectious and non-communicable diseases in the territory of the entire province. 

### 2.3. ASD Healthcare Services in the Province of Lecce

ASL/LE has developed, as part of its services, a specific healthcare assistance network for children with ASD managed by the Unit of Childhood Neuropsychiatry (UCN). This network is spread over the entire territory of the province of Lecce and is organized into 4 Territorial Poles for ASD (Figure 1): Pole 1—Lecce; Pole 2—Nardò; Pole 3—Maglie; Pole 4—Gagliano. Each pole is equipped with specialized personnel (neuropsychiatrists, psychologists, social assistants, rehabilitation therapists, speech therapists, educators) and has the task of making the diagnosis of ASD and providing medical as well as rehabilitative assistance to children with ASD, including psychological support to the families.

The ASD surveillance system adopted in the province of Lecce is the same implemented in any part of Italy (the complete flow is depicted in Figure 2), and it has been carefully monitored within the EPI-ASD project. The suspected cases of ASD are reported by the various local health professionals (pediatricians, family doctors, specialists, etc.) to the territorial poles activated under the UCN, in order to get a proper diagnosis. The case definition is based on the DSM-5 criteria [13]. In the diagnostic phase and the elaboration of the therapeutic-rehabilitative project (according to the criteria established by the Regional Regulation no.9 of 8 July 2016), the UCN operates through the Territorial Centers for Autism (CAT). All the data concerning the confirmed cases of ASD are archived by the UCN. 

### 2.4. Data from the Provincial School Office of Lecce

As above described, Italian laws guarantee the right to education for children with disabilities or emotional/behavioral difficulties, including those with ASD, from up to the age of 18 providing adequate support such as a dedicated teacher. These children, after having received an official diagnosis including the level of disability, are registered in a database owned by the Provincial School Office. 

Therefore, additional data concerning 3–18-year-old children needing school support due to ASD, who attended kindergarten, primary, or secondary school in the school year 2020–2021, were taken from the database of the Provincial School Office. These data were available grouped by type of school and consequently, according to the Italian school organization, by age groups of 3–5 years, 6–11 years, and 12–18 years respectively. Furthermore, to preserve the privacy of children, this data was provided without mentioning their gender and place of residence.

### 2.5. Study Population

The target population was consisting of 109,681 children from 3 to 18 years old (Table 1) living in the province of Lecce (data provided by the National Institute for Statistics, ISTAT 1 January 2020). This population was divided into three age groups (3–5, 6–11, and 12–18 years old) corresponding to the school levels attended by the children (kindergarten, primary school, and secondary school). The 3–5 age group included 17,208 children, the 6–11 age group 40,073 children, and the 12–18 age group 52,400 adolescents.

As shown in Table 1, Pole 1—which includes Lecce (the biggest city in the province)—has the greatest number of inhabitants while Pole 4, which includes the Southern area of the province, is the least inhabited.

### 2.6. Data Analysis

In this study, the data concerning ASD diagnoses carried out up to 27 October 2020 in children and adolescents aged 3–18 years old living in the province of Lecce were collected from each UCN pole. Additionally, data about children needing school support for ASD in the school year 2020–2021 were taken from the school office of the province of Lecce. All data were entered into a Microsoft Excel database and statistically processed using MedCalc Software version 14.8.1 (MedCalc Software bvba, Ostend, Belgium).

The prevalence for both databases was calculated by taking into consideration the population aged 3–18 years living in the province of Lecce according to the dataset of the Italian National Institute of Statistics. The 95% confidence interval (CI) in each group was also calculated.

The chi-square test was used to detect any difference in the distribution of prevalence between both databases as well as groups (gender, age groups, poles). The differences were considered significant when *p* < 0.05. The complete flow chart of the methodological approach used in this study is depicted in Figure 3.

#### Ethical Aspects

The study was conducted in accordance with Helsinki Declaration and approved by the Ethical Committee of the ASL/LE with report no. 50 of 29 July 2020. All the data were collected and processed confidentially, in accordance with Italian Legislation on the protection of personal data, for research purposes.

## 3. Results

Data provided by the UCN highlighted that, by the end of October 2020, in the province of Lecce there were 509 cases of ASD among children aged from 3 to 18 years old, corresponding to a prevalence rate of 0.46% (95% CI = 0.39–0.56%). A total of 408 (0.72%; 95% CI = 0.61–0.83%) were boys and 101 (0.19%; 95% CI = 0.16–0.22%) were girls, with a male: female ratio of about 4:1. Table 2 resumes the prevalence of ASD by age and geographical area. In relation with their age, 155 ASD cases (0.9%; 95% CI = 0.39–1.08%) were diagnosed in the 3–5 years old age group, thus showing the highest prevalence, while 222 (0.55%; 95% CI = 0.47–0.70%) ASD diagnoses were confirmed in the age group 6–11 and 132 (0.25%; 95% CI = 0.21–0.30%) between 12 and 18 years old. According to the geographical distribution, pole 3 showed the highest overall prevalence (0.56%), with age-specific rates of 1.14% in children aged 3–5 years old, 0.69 cases % in the 6–11 age group, and 0.28% in the 12–18 age group.

Data provided by the school office of Lecce province about students granted individual specific support due to ASD diagnosis officially confirmed by the ASL/LE in the school year 2020/21 highlighted a total of 410 children and adolescents aged 3–18 years old, with an overall prevalence of 0.37% (Table 3). 

Specifically, 65 ASD cases (0.37%) were in the 3–5 years old age group (attending kindergarten, which in Italy is not compulsory), and 235 ASD cases (0.58%) were in the 6–11 years old age group (primary school), while the remaining 110 ASD students (0.21%) were belonging to the age group 12–18 years old (attending lower and upper secondary school).

Overall, the assessment of the prevalence through the number of children needing school support showed significantly lower values (*p* = 0.001) than the direct method based on medical records, mainly due to the underestimation of ASD diagnoses in the youngest age group. On the other hand, the prevalence rate (0.58%) computed for the primary school age group (6–11 years old) was nearly similar to that resulting from the direct analyses of ASD diagnoses provided by the ASL/LE (0.55%). The same results were found for the age group 12–18 years old showing a prevalence rate of 0.25% vs. 0.21%, when looking at UCN and school data, respectively. In both age groups, there was no significant statistical difference between the two data sources (UCN vs. Schools).

## 4. Discussion

The present study was conducted by analyzing data from the official healthcare Network for ASD implemented in the province of Lecce by the Local Health Authority and from the Provincial School Office. The former collects information regarding ASD diagnoses in the entire territory and provides care to children and assistance to their families. In particular, the data collected for this study referred to the overall cases of ASD present in 2020 in the population of children aged 3–18 living in the province of Lecce. The latter collects data about children needing school support for disabilities or emotional/behavioral difficulties, including ASD. In particular data concerning information about children attending kindergarten, primary, and secondary schools in the school year 2020–2021 were collected and compared to those provided by the Local Health Authority.

In total, there were 509 subjects diagnosed with ASD in the population aged 3–8 years old, with a prevalence of 0.46%. The prevalence has shown a significant decrease from younger to older age groups both in the whole territory and in each pole, indicating a possible increase in attention to ASD symptoms and access to surveillance and care systems over the past few years, as well as a possible increase in the incidence of the condition, especially in younger children.

In fact, the highest values were recorded in the age group between 3 and 5 years old, up to 0.9%, while they were lower in the age groups 6–11 (0.55%) and 12–18 (0.25%). Moreover, the prevalence was found higher among boys with a male:female ratio of about 4:1. This result seems to be consistent with a recent meta-analysis performed analyzing 54 prevalence studies conducted in several countries, which reported an overall ASD prevalence of 0.39%, and a male:female ratio of 4.20 in the 0–18 year old population [19]. This latter data might suggest the adoption of a gender-based perspective in the research of ASD etiology. 

Considering the different methodologies and target populations, the prevalence data recorded in the province of Lecce seem to be not so different from those recorded in other studies conducted both in Italy and in other developed countries. The first Italian population-based prevalence study on ASD was conducted in the province of Pisa by using a multistage approach including the detection of schoolchildren needing support for ASD [17]. This study showed a prevalence of ASD in children aged 7–9 years of about 1.15%. In Emilia Romagna, the data from the regional mental health services highlighted a prevalence of about 0.6% in the year 2019 among the pediatric population aged 0–17 years old, with an increase in the last few years [18].

In the United States, the prevalence rate of ASD assessed by the Autism and Developmental Disabilities Monitoring (ADDM) system increased from 0.66% to 2.27% in the 8-year-old pediatric population from 2000 to 2018 [7,8,9,10]. A Canadian study on the local prevalence of ASD reported a prevalence of 1.06% among children 5 to 17 years using school board records within a single region [12]. A South Korean population-based study among children aged 7 to 12 years old used standardized questionnaires and diagnostic procedures reporting a prevalence of 2.66% [20]. A Swedish study in 2015 showed a prevalence of ASD of 1.74% among children aged 6–12 years old [13]. 

In the six member states of the Gulf Cooperation Council (the Kingdom of Bahrain, State of Kuwait, Sultanate of Oman, State of Qatar, Kingdom of Saudi Arabia, and United Arab Emirates), the estimated prevalence of Autism was variable and ranged from 0.014% to 0.60% [21]. Finally, the prevalence rate in India ranged from 0.15% to 0.23% [22]. The different methodologies and the variability of the populations do not allow for an accurate comparison of the above results and suggest the need to adopt shared surveillance systems in order to highlight any differences not attributable to methodological bias.

From a spatial point of view, our study highlighted a different distribution of ASD prevalence over the province of Lecce. Pole 3 reported the highest overall prevalence (0.56%) and, in particular, the highest prevalence relating to the age groups 3–5 (1.14%) and 6–11 (0.69%). The reason for this geographical difference should be deeply investigated in further studies.

The comparison between the data collected through the Healthcare Network for ASD and the data collected by the School Office of Lecce province concerning schoolchildren needing support for ASD showed that this latter methodology underestimates the prevalence in the age group 3–5 years old, as in Italy it is not compulsory to attend the school before 6 years old (first class of primary school). On the other hand, the prevalence of ASD in the age group 6–11 and 12–18 years old was all in all consistent with the diagnoses directly provided by the UCN with some minor distortions, probably due to the fact that the ages of children needing support for ASD in some cases may not coincide with the ages formally established for the attended school. 

Both methods showed some advantages and disadvantages. The data included in the UCN databases are derived from direct diagnoses carried out by the UCN of the Lecce ASL. They were available by age and gender, and allowed us to evaluate the geographical distribution according to the residence of the subjects. However, they may not include cases diagnosed and followed up by health services located outside the province of Lecce. Instead, the data included in the Provincial School Office database were easily available, and for the age groups including children attending compulsory school, they provided sufficiently comprehensive general information. On the contrary, they seemed to be not adequate for calculating a reliable prevalence for the age groups including children not attending compulsory school (i.e., 3–5 years).

Nevertheless, the methodological approach used in this study should be considered as being “conservative” and able to exclude the risk of producing an overestimation of ASD prevalence. Actually, this methodology can underestimate the phenomenon because it is not able to include high-functioning autism (HFA) forms, which occur when a child does not present intellectual disabilities, but only deficits in communication, emotional expression, and social interactions or obsessive symptoms. These HFA conditions are usually diagnosed in adolescence and do not require specific support at school. 

## 5. Conclusions

In the province of Lecce, the data collected through the Healthcare Assistance Network for ASD showed an overall prevalence of 0.46% in the 3–18 year old population. The prevalence was higher in the age group 3–5 years old. Data provided by the School Office of Lecce province were comparable to those made available by the UCN for the age groups 6–11 and 12–18, thus suggesting a possible use for epidemiological purposes limited to schoolchildren and adolescents. For the youngest age group (3–5 years old), the surveillance system adopted by the Local Health Authority of Lecce was able to also detect cases that were not detected by the school office. Furthermore, ASD diagnoses were found to be distributed differently across the province. Specific studies are needed to verify the presence of genetic or epigenetic factors that could explain the geographical differences. Our work filled an information gap regarding the prevalence of ASD and provided useful information for the design of a national surveillance system for ASD through shared methodologies.

## Figures and Tables

**Figure 1 ijerph-19-06334-f001:**
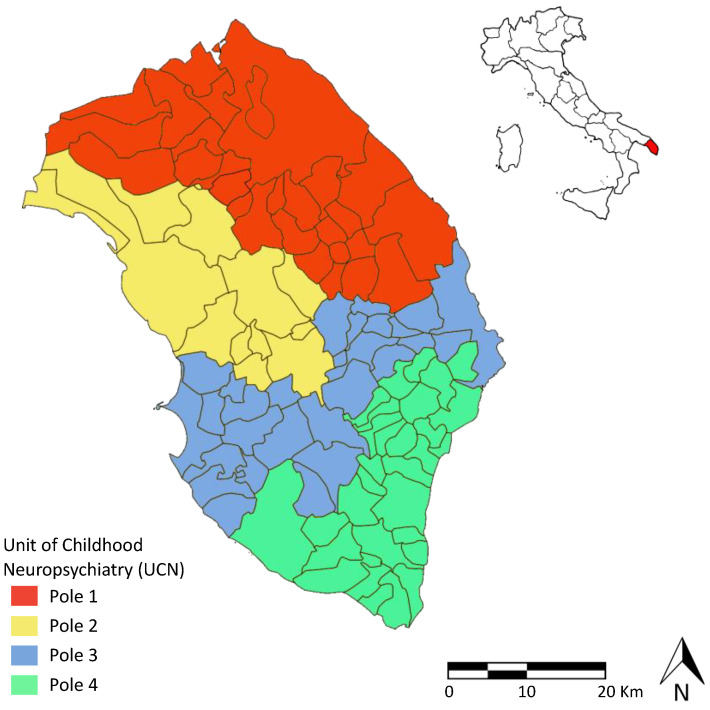
The province of Lecce and the four territorial Poles for which the Unit of Childhood Neuropsychiatry provides social and healthcare services to subjects with ASD and their families.

**Figure 2 ijerph-19-06334-f002:**
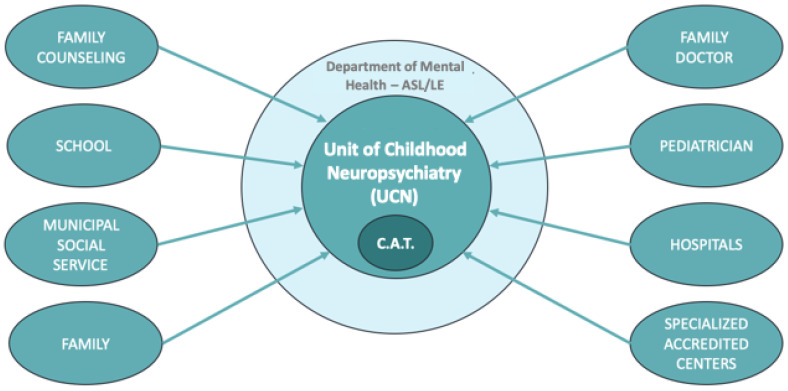
The ASD surveillance system adopted in the province of Lecce.

**Figure 3 ijerph-19-06334-f003:**
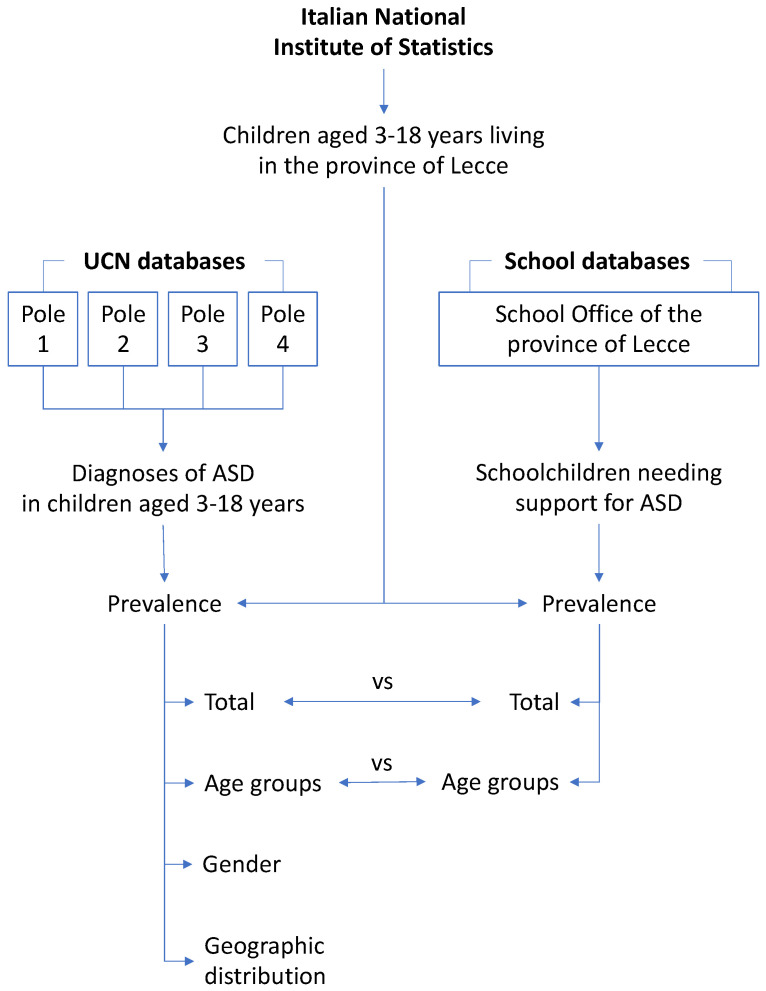
Flow chart of the methodological approach used in this study.

**Table 1 ijerph-19-06334-t001:** The target population of each territorial Pole for ASD social and healthcare services.

Age Groups	Pole 1	Pole 2	Pole 3	Pole 4	Province of Lecce
3–5 years	6875	3398	4349	2586	17,208
6–11 years	16,251	7739	10,155	5928	40,073
12–18 years	20,038	10,371	13,538	8453	52,400
Total	43,164	21,508	28,042	16,967	109,681

**Table 2 ijerph-19-06334-t002:** ASD prevalence among children aged 3 to 18 years old in the province of Lecce.

	3–5 Years		6–11 Years		12–18 Years		Total		*p*-Value
	Cases (*n*)	Prevalence (%)	Cases (*n*)	Prevalence (%)	Cases (*n*)	Prevalence (%)	Cases (*n*)	Prevalence (%)	
Pole 1	59	0.86	69	0.42	46	0.23	174	0.40	<0.0001
Pole 2	23	0.68	44	0.57	22	0.21	89	0.41	<0.0001
Pole 3	50 *	1.14	70 *	0.69	38	0.28	158 *	0.56	<0.0001
Pole 4	23	0.89	39	0.66	26	0.31	88	0.53	0.0003
Province of Lecce	155	0.90	222	0.55	132	0.25	509	0.46	<0.0001
Male	118	1.33	174	0.84	116	0.43	408	0.72	<0.0001
Female	37	0.44	48	0.25	16	0.06	101	0.19	<0.0001

* Prevalence of ASD significantly different (*p* < 0.05) among poles as evaluated by chi-square test.

**Table 3 ijerph-19-06334-t003:** Prevalence rate based on data provided by the School Office of Lecce province concerning children and adolescents needing school support for ASD in the year 2020/21.

Age Groups	Population * (*n*)	Children Needing School Support for ASD (*n*)	Prevalence Rate (%)
3–5 years	17,208	65	0.37
6–11 years	40,073	235	0.58
12–18 years	52,400	110	0.21
Total	109,681	410	0.37

* Residents provided by ISTAT.

## Data Availability

Data supporting reported results can be found at the Laboratory of Hygiene, Department of Biological and Environmental Science and Technology, University of Salento, Centro Ecotekne, Via Monteroni 165, 73100—Lecce (LE), Italy.

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
