# Peer review of "A Feasible Methodological Approach to Estimate the Burden of Autism Spectrum Disorder: Results from the EPI-ASD Study in the Province of Lecce (Southern Italy)"

_ijerph, 2022, doi:10.3390/ijerph19106334_

Round 1
Reviewer 1 Report
The paper is accurate and aiming to fill a gap about ASD prevalence. still, I have some suggestions in order to get to improve the manuscript.
1 I would mention the overall project in the study design section, but I would not refer to possible environmental factors increasing the prevalence of ASD from the other parts. This is a quite delicate topic that need a thorough discussion instead of a quick mention. There are indeed several papers suggesting that no environmental factors are implicated (Brugha et al, 2011).
2 the description of the referral system for ASD caseness is quite accurate. Still, some details are unnecessary and make the reading very hard (e.g. naming every social health district is not really useful; describing the different poles is unnecessary as well, as very few comments are made in the discussion about this).
3 in the results: I personally prefer not to repeat in the results section data that are already shown in the tables. Moreover, I would offer authors to express data as percent, instead of per thousand for quicker comparison.
4 what p. values in table 2 are referring to?
please find further details in the document attached

Author Response
Answer to Reviewer 1
The paper is accurate and aiming to fill a gap about ASD prevalence. still, I have some suggestions in order to get to improve the manuscript.
We thank the Reviewer for his valuable comments which are helpful for improving the quality of the manuscript. Below we have indicated point by point the revisions we have made.
1 I would mention the overall project in the study design section, but I would not refer to possible environmental factors increasing the prevalence of ASD from the other parts. This is a quite delicate topic that need a thorough discussion instead of a quick mention. There are indeed several papers suggesting that no environmental factors are implicated (Brugha et al, 2011).
We thank the Reviewer for the suggestion. In the study design section, we have deleted the reference to the possible involvement of environmental factors in the development of ASD, highlighting, instead, that one of the objectives of the project was to investigate possible associations between ASD and individual or family factors. Moreover, any reference to environmental or epigenetic factors in the development of ASD have been removed as they were not relevant for the purpose of the study.
2 the description of the referral system for ASD caseness is quite accurate. Still, some details are unnecessary and make the reading very hard (e.g. naming every social health district is not really useful; describing the different poles is unnecessary as well, as very few comments are made in the discussion about this).
As suggested, we have deleted the unnecessary details.
3 in the results: I personally prefer not to repeat in the results section data that are already shown in the tables. Moreover, I would offer authors to express data as percent, instead of per thousand for quicker comparison.
We thank the reviewer for the valuable comment. We have included in the comments only the relevant data deleting the other data reported in the table 2. Moreover, we have expressed data as percent, instead of per thousand throughout the manuscript, as suggested.
4 what p. values in table 2 are referring to?
We have described in the caption of Table 2 what p-value referrers to.
please find further details in the document attached
We thank the reviewer for the valuable comment. We have replaced the citation no. 15 with a more recent one, which reported the results of a meta-analysis performed on 54 prevalence studies and a male:female ratio referred to a 0-18 years old population. In this way our results could be comparable.

Reviewer 2 Report
I would like to thank the authors of this paper for their work.
In general, I think they do a good overview of the background for the study, however, data in line 51 should be updated, as there is a new reference (Maenner MJ, Shaw KA, Bakian AV, et al. Prevalence and Characteristics of Autism Spectrum Disorder Among Children Aged 8 Years — Autism and Developmental Disabilities Monitoring Network, 11 Sites, United States, 2018. MMWR Surveill Summ 2021;70(No. SS-11):1–16. DOI: http://dx.doi.org/10.15585/mmwr.ss7011a1external icon) that reports an even higher prevalence: “one in 44 children aged 8 years was estimated to have ASD.” Also, the introduction stated many genetic-based studies that weren’t really relevant to the study.
Regarding the methods section, data analyses were not explained, and I think the paper would benefit if the authors added a subsection in the methods part explaining how they have estimated prevalence or they have compared data between the different databases. Also, prevalence data would be better interpreted if they reported confidence intervals (CIs). Figures 1 and 2 could be synthesized into just one figure, where all the information can be seen in just one look.
When presenting the results, I found the information confusing due to not having a previous explanation of data analyses and the specific objectives of the study. For example, I am not sure what the p-value on the number of cases in Table 2 means. In lines 183-186, the information reported is not presented before. In lines 195-199, the concept of incidence and results regarding this issue are introduced for the first time, without previous background in the introduction or in the aims of the study. If incidence is also of interest to the authors, they should present this subject within the introduction and aims, otherwise, I think the paper would be clear if they removed this part completely from the results and discussion sections.
Information about prevalence reported in the discussion section seems a bit reiterative and similar to what has been said in the introduction section. I believe this paper will benefit from a deeper argumentation regarding the results obtained. Lines 259-260 “This finding may be due to genetic, epigenetic or environmental differences that need to be deeply investigated in further studies.” are too general and the authors do not give any arguments explaining this suggestion. Many other factors could be influencing these results and they would be equally valid, so I would avoid overgeneralizations. Also, I think there should be some sort of suggestion of reasons for more students in the system receiving support than those that are officially diagnosed in the region for the 6-11 and 12-18 age groups. Additionally, the discussion will enrich if the authors present some advantages and disadvantages of each method.
Author Response
Answer to Reviewer 2
I would like to thank the authors of this paper for their work.
In general, I think they do a good overview of the background for the study, however, data in line 51 should be updated, as there is a new reference (Maenner MJ, Shaw KA, Bakian AV, et al. Prevalence and Characteristics of Autism Spectrum Disorder Among Children Aged 8 Years — Autism and Developmental Disabilities Monitoring Network, 11 Sites, United States, 2018. MMWR Surveill Summ 2021;70(No. SS-11):1–16. DOI: http://dx.doi.org/10.15585/mmwr.ss7011a1external icon) that reports an even higher prevalence: “one in 44 children aged 8 years was estimated to have ASD.”
We thank the reviewer for the very valuable comment. We added the suggested reference with the new data about prevalence of ASD.
Also, the introduction stated many genetic-based studies that weren’t really relevant to the study.
The references to genetic factors in the development of ASD have been removed from the introduction as they were not relevant for the purpose of the study.
Regarding the methods section, data analyses were not explained, and I think the paper would benefit if the authors added a subsection in the methods part explaining how they have estimated prevalence or they have compared data between the different databases. Also, prevalence data would be better interpreted if they reported confidence intervals (CIs).
We thank the reviewer for the suggestion. In the methods section we have added the data analysis subsection in which we have explained how prevalence was estimated and compared among groups. In addition, the 95% confidence interval was reported for the overall prevalence and for prevalence among gender and age groups.
Figures 1 and 2 could be synthesized into just one figure, where all the information can be seen in just one look.
Figures 1 and 2 were synthetized in just one figure as the reviewer suggested removing all the unnecessary information.
When presenting the results, I found the information confusing due to not having a previous explanation of data analyses and the specific objectives of the study.
As previously suggested, in the methods section we have added the data analysis subsection
For example, I am not sure what the p-value on the number of cases in Table 2 means.
We have described in the caption of Table 2 what p-value referrers to.
In lines 183-186, the information reported is not presented before.
The comment about the differences between the two databases was moved in discussion section.
In lines 195-199, the concept of incidence and results regarding this issue are introduced for the first time, without previous background in the introduction or in the aims of the study. If incidence is also of interest to the authors, they should present this subject within the introduction and aims, otherwise, I think the paper would be clear if they removed this part completely from the results and discussion sections.
We thank the reviewer for the valuable comment. We completely removed the discussion about the concept of incidence both from result and discussion section.
Information about prevalence reported in the discussion section seems a bit reiterative and similar to what has been said in the introduction section.
As suggested, we have deleted the reiterative information in discussion section, as they were reported in the introduction section.
I believe this paper will benefit from a deeper argumentation regarding the results obtained.
We thank the reviewer for the suggestion. We have discussed more deeply our results. In particular we added more recent literature, our consideration about the different methodologies adopted in the various studies, and some comments regarding the advantages and disadvantages of both methods.
Lines 259-260 “This finding may be due to genetic, epigenetic or environmental differences that need to be deeply investigated in further studies.” are too general and the authors do not give any arguments explaining this suggestion. Many other factors could be influencing these results and they would be equally valid, so I would avoid overgeneralizations.
We thank the reviewer for the valuable suggestion. The reference to environmental or epigenetic factors in the development of ASD have been removed as they were not exhaustive and not relevant for the purpose of the study. Instead, we have highlighted the need to deeply investigated the different geographical distribution of prevalence to better understand the reason of this epidemiological phenomenon.
Also, I think there should be some sort of suggestion of reasons for more students in the system receiving support than those that are officially diagnosed in the region for the 6-11 and 12-18 age groups.
As suggested, we have given some reasons for the distortion observed in the number of “cases” detected in the two systems.
Additionally, the discussion will enrich if the authors present some advantages and disadvantages of each method.
We have added some comments regarding the advantages and disadvantages of both methods.

Reviewer 3 Report
The manuscript entitled “A Feasible Methodological Approach to Estimate the Burden of Autism Spectrum Disorder: Results from the EPI-ASD Study in the Province of Lecce (Southern Italy)” By Giovanni Imbriani and colleagues is simple and interesting.
Major concern is the authors didn’t provide the gender based clearly for the small samples with ASD.
Authors shall described the diagnosis criteria, statement “, where all the subjects with ASD living in the province are diagnosed and followed up” is incomplete. Are the collected subjects fulfilled the DSM-5 criteria for autism?
I am wondering the reason for grouping “This population was divided into three age groups (3-5, 6-11, and 12-18 years old) corresponding to the school levels attended by the children (kinder- garten, primary school and secondary school).” Is there any criteria considered for grouping them? If not, factors considered to divided into three age groups (3-5, 6-11, and 12-18 years old) shall be described appropriately in the methodology.
Authors have discussed only the data from certain region of the world. They shall discussed the data also from other regions (Qoronfleh, M.W., Essa, M.M., Alharahsheh, S.T., Al-Farsi, Y.M. and Al-Adawi, S., 2019. Autism in the Gulf states: A regional overview. Frontiers in Bioscience, Landmark, 24, pp.324-336.
Patra, S. and Kar, S.K., 2021. Autism spectrum disorder in India: a scoping review. International Review of Psychiatry, 33(1-2), pp.81-112.).A column of actual p value should be added in the “Table 2.” This will ensure the clarity on the ASD prevalence among children aged 3 to 18 years old in the province of Lecce.
Methodology followed for the ASD diagnosis and officially confirmation shall be elaborated to support the statement “specific support due to ASD diagnosis officially confirmed by the ASL/LE”.
The data provided in the Table 2 and Table 3 also in Table 4 should be elaborated with gender distribution, this will provide depth idea.
Detail flowchart of the “methodological approach used in this study” shall be provided for better understanding with complete details.
How do the authors expect the school authorities to diagnose the ASD children in the school? The details on this can give better clarity on the statement “For the youngest age group (3-5 years old), the surveil- lance system adopted by the Local Health Authority of Lecce was able to detect also cases that were not detected by the school office.”.
Generalised conclusions shall be replaced with specific conclusions from the study.
Author Response
Answer to Reviewer 3
The manuscript entitled “A Feasible Methodological Approach to Estimate the Burden of Autism Spectrum Disorder: Results from the EPI-ASD Study in the Province of Lecce (Southern Italy)” By Giovanni Imbriani and colleagues is simple and interesting.
Major concern is the authors didn’t provide the gender based clearly for the small samples with ASD.
We thank the reviewer for the valuable comment. We added more data about gender of children and adolescents with ASD (number and prevalence) according to their age. The data have been reported in table 2.
Authors shall described the diagnosis criteria, statement “, where all the subjects with ASD living in the province are diagnosed and followed up” is incomplete. Are the collected subjects fulfilled the DSM-5 criteria for autism?
Thanks for the suggestion. In the introduction section we have added the criteria (DSM-5 criteria) by which ASD is diagnosed in the Italian CAT centers. These criteria were also specified in the method section.
I am wondering the reason for grouping “This population was divided into three age groups (3-5, 6-11, and 12-18 years old) corresponding to the school levels attended by the children (kinder- garten, primary school and secondary school).” Is there any criteria considered for grouping them? If not, factors considered to divided into three age groups (3-5, 6-11, and 12-18 years old) shall be described appropriately in the methodology.
The children population was divided into three age groups (3-5, 6-11, 12-18 years-old) in order to compare the prevalence between results from ASL dataset and school dataset since for the latter we had data regarding the overall children attending the various school levels.
Authors have discussed only the data from certain region of the world. They shall discussed the data also from other regions
(Qoronfleh, M.W., Essa, M.M., Alharahsheh, S.T., Al-Farsi, Y.M. and Al-Adawi, S., 2019. Autism in the Gulf states: A regional overview. Frontiers in Bioscience, Landmark, 24, pp.324-336.
Patra, S. and Kar, S.K., 2021. Autism spectrum disorder in India: a scoping review. International Review of Psychiatry, 33(1-2), pp.81-112.).
We thank the reviewer for the valuable comment. We added results from other world regions, as suggested.
A column of actual p value should be added in the “Table 2.” This will ensure the clarity on the ASD prevalence among children aged 3 to 18 years old in the province of Lecce.
We have added a column of actual p value in the Table 2.
Methodology followed for the ASD diagnosis and officially confirmation shall be elaborated to support the statement “specific support due to ASD diagnosis officially confirmed by the ASL/LE”.
A section about the data provided by the Provincial School Office of Lecce was added to describe how data were collected.
The data provided in the Table 2 and Table 3 also in Table 4 should be elaborated with gender distribution, this will provide depth idea.
We thank the reviewer. We have added in table 2 the gender distribution. Unfortunately, this hasn’t been possible for table 3 because data were not available by gender to preserve the privacy of children.
Detail flowchart of the “methodological approach used in this study” shall be provided for better understanding with complete details.
A detailed flowchart of the methodological approach used in this study has been added in the method section.
How do the authors expect the school authorities to diagnose the ASD children in the school? The details on this can give better clarity on the statement “For the youngest age group (3-5 years old), the surveil- lance system adopted by the Local Health Authority of Lecce was able to detect also cases that were not detected by the school office.”.
A section about the data provided by the Provincial School Office of Lecce was added to describe the details about the data collection.
Generalised conclusions shall be replaced with specific conclusions from the study.
We thank the reviewer for the suggestion. We have discussed more deeply our results. In particular we added some comments regarding the advantages and disadvantages of both methods and possible explanation of some results.

Round 2
Reviewer 3 Report
Revised MS can be accepted